# Metabolic Syndrome in Obese Children—Clinical Prevalence and Risk Factors

**DOI:** 10.3390/ijerph18031060

**Published:** 2021-01-25

**Authors:** Agnieszka Jankowska, Michał Brzeziński, Anna Romanowicz-Sołtyszewska, Agnieszka Szlagatys-Sidorkiewicz

**Affiliations:** 1Department of Paediatrics, Gastroenterology, Allergology and Paediatric Nutrition, Medical University of Gdańsk, 80-462 Gdańsk, Poland; michal.brzezinski@gumed.edu.pl (M.B.); aga1@gumed.edu.pl (A.S.-S.); 2Department of Paediatric Cardiac Surgery in Gdańsk, Medical University of Gdańsk, 80-846 Gdańsk, Poland; aneczka.romanowicz@wp.pl

**Keywords:** metabolic syndrome, risk factors, obesity children

## Abstract

The prevalence of childhood obesity is increasing worldwide. Some obese children can go on to develop metabolic syndrome (MetS), but exactly who among them remains to be determined. The aim of this study was to indicate predisposing factors for metabolic syndrome, especially those that can be modified. The study comprised 591 obese children aged 10–12 years. They were all Caucasian residents of Gdańsk, Poland, with similar demographic backgrounds. Clinical examination, anthropometry, biometric impedance analysis, blood tests (including oral glucose tolerance tests (OGTT) and insulinemia), and dietary and physical activity evaluation were conducted. The results of our study show that the risk factors for MetS or any of its components include male sex, parental (especially paternal) obesity, low body mass at birth, as well as omitting breakfast or dinner. There are few risk factors for metabolic syndrome both in obese adults and children. Some of these predictors can be modified, especially those in relation to lifestyle. Identifying and then influencing these factors may help to reduce the development of metabolic syndrome and consequently improve health and quality of life.

## 1. Introduction

The prevalence of obesity is rapidly increasing worldwide. According to the World Health Organization (WHO), in 2016 obesity affected as many as 650 million people, and two billion adults were considered overweight (i.e., at a high risk of becoming obese); 41 million children under 5 years of age were overweight or obese and over 340 million children and adolescents aged 5–19 were overweight or obese [1]. The number of children with obesity continues to increase and it is therefore expected that complications from obesity in these age groups will also increase.

Obesity, defined as the excess of fat tissue, is wide spread all over the world and it affects all age groups [2]. Obesity, among other factors, is well known risk factor for metabolic syndrome (MetS) which comprises cardiovascular diseases (CVDs), diabetes mellitus (DM), hypertension, and atherosclerosis, as well as other complications [3]. Although the etiology of obesity is complex, genetic predisposition is permissive and actually interacts with environmental agents including physical activity and diet. Heritable factors seem to make a 40–85% contribution to obesity’s etiology. Apart from genetic predisposition, other recognized constituents such as the metabolom and metabolic programming during both the gestational and post-gestational periods can be modified to some extent [4]. Consequently, development of obesity and its complications might be reduced [5]. 

Metabolic syndrome is a recognized consequence of obesity and it can occur as early as in adolescence. Certainly, not all obese children will develop all or any of the complications of obesity. Which children living with obesity are most prone to MetS remains to be fully elucidated. 

In a systematic review of 85 studies in children, the median prevalence of metabolic syndrome in all populations was 3.3% (range 0–19.2%), in overweight children it was 11.9% (range 2.8–29.3%), and in obese subjects it was29.2% (range 10–66%). For non-obese, non-overweight populations, the range was 0–1% [6]. Almost 90% of obese children and adolescents have at least one feature of metabolic syndrome [7]. On the basis of National Health and Nutrition Examination Survey (NHANES) 1999 to 2002 data, the prevalence of metabolic syndrome in adolescents 12 to 19 years old ranged from 0 to 9.4%; variation in this estimate was the result of different criteria used to define metabolic syndrome [8]. In a report from (Biobank Standardisation and Harmonisation for Research Excellence in the European Union (BIOSHARE-EU), prevalence of metabolic syndrome in obese subjects ranged from 24 to 65% in females and from ≈43 to ≈78% in males and substantially exceeded the prevalence in metabolically healthy obese subjects [9]. These divergences depended on the country.

In the light of children’s dynamic growth and maturation, the population of children is unique. Obesity is diagnosed on the basis of centile charts, when BMI ≥95th centile. Moreover, according to The International Diabetes Federation (IDF) consensus—commonly used by most of the authors—metabolic syndrome may be recognized in children not younger than 10 years old [10]. It takes some time before metabolic syndrome complicates obesity. Nevertheless, the younger the child becomes obese, the earlier in life he or she might suffer from its complications [11]. However, one must remember that not all children suffering from obesity will develop MetS. 

The aim of our study was to identify factors favoring the presence of one or all compounds of metabolic syndrome in obese children. Secondly, we analyzed the results in order to indicate which of them are modifiable. 

## 2. Materials and Methods

This study was a part of the “6–10–14 for Health” integrated weight management program for children with overweight and obesity from Gdansk, Poland. The detailed design is available in previous publications and on request [12,13,14]. 

The analyzed data included children aged 10–12 years attending the intervention program in 2011–2015. Children were screened in primary schools and if overweight or obesity was diagnosed they were invited to the multidisciplinary, 12-month-long program. Patient flow is presented in Figure 1. 

The demographic background of the participants was similar in the terms of ethnicity and socioeconomic status, however some minor discrepancies were present. The population of the City of Gdansk is over 99% Caucasian.

All the children were examined by pediatrician during the first intervention visit (including weight, height, and waist circumference measurement) and bioelectric impedance analysis (BIA) was performed. All the children were referred for blood tests within 4 weeks of the first visit.

Medical history was taken and collected data included body mass at birth (below 2.5 kg was assumed as hypotrophy, more than 4.0 kg—macrosomy), parents’ BMI, any metabolic disease in family members, sleeping disorders, and gastrointestinal complaints. 

Body mass, BMI centile, waist circumference centiles, and blood pressure centiles were assessed using Polish centile charts, as recommended by the WHO [15,16,17]. Overweight was diagnosed if the BMI centile was ≥85th and obesity was diagnosed when BMI centile was ≥95th on the recommended centile charts [15]. 

Waist circumference (WC) over the 90th percentile and waist–hip ratio (WHR) > 0.8 for girls and >0.9 for boys were interpreted as abnormal. 

For children younger than 10 years old, waist–hip ratio (WhtR) seems to be more accurate than WC. Abnormal WhtR was defined when WhtR ≥0.5.

In our study, although all the participants were at least 10 years old, WhtR was also calculated.

The results of biometric impedance analysis were assessed according to standard values [18]. 

Lifestyle was evaluated on the basis of information obtained during the visit. 

Physical activity was evaluated by means of a Kash Pulse Recovery Test [19,20,21] and classified as excellent, very good, good, moderate, poor, and very poor.

Nutritional habits were evaluated by a dietitian on the basis of recall data given by children and parents. Specific questions regarding quantity and quality of meals constituted the original questionnaire. Special attention was paid to breakfast (first course within 2 h after waking up) and dinner (last meal eaten 2 h before sleep), including the time of day.

Blood tests comprised aminotransferases, lipid fractions, thyroid stimulating hormone, tyrosine, Hb1c, and oral glucose tolerance test (OGTTs), along with insulin levels at the same time points, and the results were compared to standard values for appropriate age and sex.

Metabolic syndrome (MetS) was diagnosed according to IDF [22] in children with WC > 90th centile and at least two out of the following metabolic features: HDL < 40 mg/dL, TG > 150 mg/dL, glycemia > 100 mg/dL, and blood pressure ≥ 130/85 mmHg.

Statistical analyses included normal distribution of continuous variables, which was verified with the Shapiro–Wilk test [23]. Descriptive statistics are presented as the mean or median and standard deviation from the mean. Between-group comparisons were carried out using the Mann–Whitney U test and ANOVA Kruskal–Wallis test [23]. Nonparametric tests were chosen because of the large number of significant Shapiro tests, which were used for normality assumption assessment. All statistical tests were 2-tailed and performed at the 5% level of significance. Statistical analysis was performed using Statistica 10 software (TIBCO Software Inc., Tulsa, OK, USA 2014). 

This study was accepted by the Independent Bioethics Committee for Scientific Research at the Medical University of Gdańsk (NKBBN/228/2012) on 25 June 2012. The study is registered in clinicaltrials.gov (NCT number: NCT04143074).

## 3. Results

591 children aged 10–12 years who entered the program, completed the questionnaire and had blood tests performed. None of the children had any chronic disease that could influence investigated parameters as well as no infection on the day of examination.

The girls were younger (*p* = 0.031), shorter (*p* = 0.028), and had lower WC (*p* < 0.0001), lower WHR (*p* < 0.0001), lower WhtR (p = 0.002), and lower DBP (*p* = 0.044) compared to the boys. Moreover, higher BMI centile (*p* < 0.001) and higher fat tissue content were characteristic for girls. Elevated systolic blood pressure (SBP) was found in 10% of children, with no difference between girls and boys (Table 1).

Based on BMI percentile criterion there were 401 (67.9%) overweight (BMI 85–95 centile) and 190 (32.1%) obese children (BMI ≥ 95 centile). Obesity was diagnosed in 85 (30.9%) girls and in 105 (33.2%) boys, whereas 190 (69.1%) girls and 211 (66.8%) boys were overweight. Girls had higher BMI centiles compared to boys (*p* < 0.0001).

Among 76 (12.9%) children with metabolic syndrome there were 30 girls and 46 boys (Table 2 and Table 3).

73% of children with MetS had elevated blood pressure in comparison with 1.1% of children from groups II and III (*p* = 0.005).

Reduced levels of HDL (<40 mg/dL) were found in all the children with MetS and in 27.6% of the remaining children (i.e., groups II and III), which was significantly different (*p* = 0.029). More than half of participants with MetS (55.3%) had increased TG concentration compared to 4% of the remaining children (*p* = 0.001).

### 3.1. Analysis of Prevalence of Proposed Risk Factors of MetS

To achieve this aim we compared children constituting group I (MetS+) to the rest of the children i.e., groups II and III (Met− and MetS+/−).

According to medical history, hypotrophy at birth was three times more prevalent in boys with MetS than in boys without MetS. Discerning gender showed that low birth weight was more prevalent in boys with MetS (*p* = 0.034). All the girls with MetS were eutrophic at birth.

Further analyses revealed that 47.3% of fathers of children with MetS were obese and 14.5% fathers had normal body mass. Maternal BMI did not differ between the groups.

There was no difference between the subgroups in terms of analyzed metabolic diseases (DM2, hypothyroidism, MetS, or dyslipidemia) in children’s parents. Similarly, prevalence of cardiovascular diseases in parents was not significant.

The correlation between metabolic syndrome and its components was significant for

Paternal obesity (*p* = 0.023);Obesity in at least one parent (*p* = 0.046);Low body mass at birth in boys (*p* = 0.046).

### 3.2. Nutrition Analysis Based on Medical History and Recall Questionnaire

The results provided that children with MetS (44.7%) omitted breakfast more often than children from group II (35%) and group III (28.7%) (*p* = 0.03). This phenomenon was especially distinctive in boys. 

Therefore, not eating breakfast was a risk factor not only for MetS (*p* = 0.027, OR = 1.74; 95%CI: 1.06–2.87) but also for at least one of its components (*p* = 0.036) (OR = 1.46; 95% CI: 1.02–2.09).

Omitting dinner was more prevalent in children from groups I and III (OR = 1.63; 95% CI: 1.13–2.35) as well as in boys (OR = 2.66; 95% CI: 1.56–4.55) (see Appendix A, Additional File S1). Not eating dinner seems to be another risk factor for MetS.

On the other hand, there were no differences in eating fruit or vegetables (in terms of quantity and frequency) between children with MetS and the rest of the study population.

### 3.3. Physical Activity

The physical performance results (evaluated by the Kash Pulse Recovery Test) are presented in Appendix A, Additional File S1. This shows that children with MetS had very poor physical performance, three times more often than the rest of the children. However, boys with MetS had significantly worse physical performance than the rest of the males (*p* = 0.15). 

The correlation between metabolic syndrome and its components was significant for

Omitting dinner: in the whole population (OR = 1.63; 95% CI: 1.13–2.35), especially in boys (OR = 2.66; 95%CI: 1.56–4.55);Not eating breakfast (*p* = 0.036) (OR = 1.46; 95% CI: 1.02–2.09).

Biochemical assessment

Elevated levels of insulin were most common in children with MetS; twice as much as in children from group II and fourfold than in group III. Significantly, more obese children without MetS had normal insulin levels (*p* = 0.001) (Table 4).

Insulin resistance was also more common in children with MetS compared to the remaining children (Table 5).

Five children with MetS had fasting glucose >126 mg/dl and were consulted by a diabetologist.

Children without MetS had normal fasting glucose concentration, whereas children from groups I and III were pathological in 27.6% cases and 9.9% cases, respectively.

Abnormal Homeostatic Model Assessment for Insulin Resistance (HOMA-IR) values were significantly more common in children with metabolic syndrome (*p* = 0.005) (Table 6).

Although children with MetS had elevated aminotransferase activity more often, the difference was significant for Alanine transaminase (ALT) only (p = 0.011). Hypertriglyceridemia, Hyperglycemia and lower HDL concentration were similar in boys and girls. (Table S1). Nevertheless, the mean values of these parameters differed: boys had higher concentration of glucose and HDL and lower TG concentration (Appendix A).

## 4. Discussion

The prevalence of obesity is high worldwide and some describe it as pandemic. This disease is known to be one of the so-called “lifestyle diseases” and it develops mostly in adolescence and adulthood. However, the incidence of obesity in children is growing fast and the burdens of its complications must be considered not only from a medical but also a socioeconomic point of view. 

Obesity increases the risk of other diseases of affluence such as hypertension, dyslipidemia, and glucose intolerance, and at the same time it is a well-known risk factor for CVD and MetS [24,25,26,27]. 

Many authors have reported that children with a BMI over the 75th centile have higher morbidity and mortality due to DM2 and CVD in adulthood [28,29,30,31]. Thus, quality and expectancy of life certainly is and will be affected by obesity. 

Metabolic syndrome has become the epitome of obesity’s complications with a high impact on human wellbeing.

Despite various definitions of MetS in children, apparently not all teenagers with obesity will develop metabolic syndrome or even one of its components. Which obese children (and in fact which of the whole obese population) are especially predisposed to MetS is still not fully understood. However, being aware of the risk factors for metabolic syndrome might allow prevention of MetS or at least minimize its prevalence and consequences.

In our study, 12.9% of obese children 9–12 years old, participating in “6–10–14 for Health” had metabolic syndrome diagnosed—10.9% of girls and 14.6% of boys. These results are similar to those in other publications [32]. It seems that prevalence of obesity and metabolic syndrome in children is more or less the same all over the world. Taking into consideration sociodemographic variables, differences in religion, economic status, etc., there must be some stand-alone background.

According to Abdullah et al., young age at the onset of obesity, as well as the time period for which obesity lasts, are essential factors for MetS in adolescence [33]. Clearly, appropriate prophylaxis should be undertaken as soon as possible to stop this process.

Our main aim was to identify children who were at the highest risk of developing MetS. Recognition of predisposing agents which can be modified may be crucial, since it is our deep belief that not undertaking any prevention will eventually lead to MetS.

The results show that obese children with metabolic syndrome are characterized by poor physical performance, bad nutrition habits, and glucose intolerance with insulin resistance.

It is hard to say whether poor physical performance is secondary to obesity and MetS, or rather the reason for these. Contemporary lifestyle with ubiquitous hi-technology, and fast, mechanized transportation, do not mobilize children to undertake physical activity. A sedentary lifestyle also favors certain nutritional choices, including fast-foods, snacks, and sweet beverages-all high-sugar, high-fat products.

Similarly to Mazur et.al., we also found that pathological WhtR predisposes children to both MetS and certain components of the syndrome [34].

In our study, we found that boys with low body mass at birth more often had MetS. Obviously this fact is irreversible, but nevertheless it is known that metabolic programming, which begins in fetal life, carries on after birth. There is a slight opportunity to influence this process by means of a healthy lifestyle for pregnant woman and this can be achieved by educating not only doctors and health providers but also mothers. Promoting breastfeeding exclusively with recommended weaning time is an easy way to influence metabolic programming and weight gain in the first 2–3 years of life.

Both or either parent’s obesity—especially the father’s—might suggest genetic predisposition to excess body mass in a child, but, on the other hand, it could illustrate the family’s lifestyle. It is already known that obese children skip breakfast more often than healthy children [35]. Contrary to some authors, we also found that omitting dinner is related to metabolic syndrome or to components of it.

This feeding pattern fits into the modern model of life, characterized additionally by little physical activity and much sedentary time (at school, work, and home). Similar observations have already been published [36].

The recall data collection is undoubtedly the limitation of this study. Both interviews and questionnaires have flaws. Respondents may have not given accurate answers. Moreover, the recall data most probably are not precise and are rather “approximate”; some information may also have been omitted.

In some cases the prospective dietary records were ordered, but could be verified only after the first interventional visit (i.e., during the 2nd or even 3rd visits, when the recommended actions should have been implemented). The strengths of this study, however, were the number of subjects and detailed lifestyle and laboratory evaluation, as well the performance of a test to assess physical fitness—these are rarely performed in population studies in children.

One cannot precisely assess what is a result of genetic involvement and what is a result of a family’s habits in the development of obesity. Nevertheless, the latter can be modified. However, if genetic predisposition runs in the family, this is a red flag and preventive measurements should be undertaken in order to minimize the unavoidable consequences of obesity and MetS. 

Sound knowledge and education on energy balance (via proper diet and physical activity) should be promoted as a driving force against obesity. Changes implemented for a child’s benefit will benefit the whole family.

To our knowledge this is the first study on the factors predisposing obese children and teenagers to metabolic syndrome that has been carried out in such a large, homogenous population (age, residency).

Taking into account genetic predisposition and environmental influences, we tried to indicate modifiable risk factors.

The results of our study could be used as warning signals for subjects who are genetically predisposed to obesity. In these children certain preventive measures should be taken. Although, unfortunately, it has become a catchphrase, physical activity and a healthy diet are the best way to lessen the risk of severe complications from obesity and metabolic syndrome itself.

## 5. Conclusions

Among the many risk factors of metabolic syndrome, besides those that are irreversible (such as body mass at birth, gender, genetic predisposition, etc.) there are many factors that are dependent on lifestyle. Proper, increased physical activity and rational nutrition (regular healthy meals) can be modified and, by this, the risk of metabolic syndrome in obese children can be diminished in an inexpensive way.

Education and preventive companies addressing health providers and parents are required in order to lessen morbidity in relation to obesity and metabolic syndrome or at least minimize their prevalence and consequences.

Intervention programs addressed to all overweight and obese children, especially those who are at risk of metabolic syndrome, should be designed (preferably on an international level).

## Figures and Tables

**Figure 1 ijerph-18-01060-f001:**
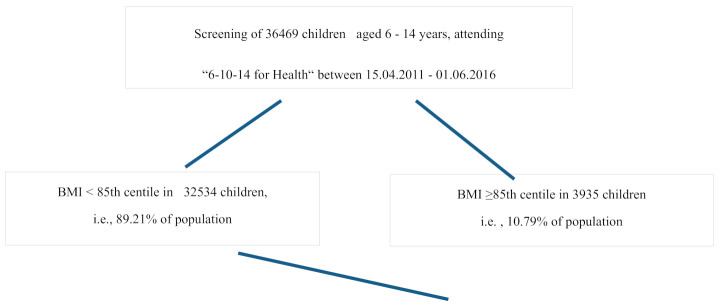
Patient screening and qualification for the study-study flow.

**Table 1 ijerph-18-01060-t001:** Results of anthropometry, blood pressure, and fat tissue content in the study population.

Girls (*n* = 275)		Boys (*n* = 316)		
**Mean**	**SD**	**mean**	**SD**	***p***
10.36	0.74	10.5	0.8	0.031
148.3	7.89	149.7	7.7	0.028
53.12	9.88	54	9.46	0.074
24.01	2.88	24.1	2.73	0.541
94.18	3.91	92.9	3.81	0.000
76.46	7.34	78.85	7.58	0.000
87.99	7.5	87.72	6.7	0.952
0.87	0.07	0.9	0.09	0.000
0.52	0.05	0.53	0.05	0.002
112.61	11.77	112.87	11.4	0.820
68.87	8.41	70.11	8.38	0.044
33.08	4.56	27.61	5.42	0.000

*p* < 0.05 Mann–Whitney U test analysis.

**Table 2 ijerph-18-01060-t002:** Number of components of metabolic syndrome in girls and boys.

	Total		Girls		Boys		*p*
**Number of Components of Metabolic Syndrome (*n*)**	***n***	**%**	***n***	**%**	***n***	**%**	
0	243	41.1	119	43.3	124	39.2	0.239
1	272	46	126	45.8	146	46.2	
2	66	11.2	24	8.7	42	13.3	
3	10	1.7	6	2.2	4	1.3	

*p* < 0.05 Chi-square test analysis by sex.

**Table 3 ijerph-18-01060-t003:** Number of girls and boys in Groups I, II and III.

Group	Total		Girls		Boys		*p*
	***n***	**%**	***n***	**%**	***n***	**%**	
MetS+	76	12.9	30	10.9	46	14.6	0.349
MetS–	243	41.1	119	43.3	124	39.2	
MetS+/–	272	46	126	45.8	146	46.2	

*p* < 0.05 Chi-square test analysis.

**Table 4 ijerph-18-01060-t004:** Fasting insulin level (Ins), one and two hours after glucose intake (oral glucose tolerance test (OGTT)) in study population.

	MetS+		MetS−		MetS+/−		MetS+ vs. MetS− vs. MetS+/−	MetS+ and MetS+/− vs. MetS−	MetS+ vs. MetS− and MetS+/−
							*p*-Value		
Fasting insulin level	*n*	%	*n*	%	*n*	%	0.001	0.001	0.001
Elevated (≥15 ng/mL)	37	48.6	32	13.2	59	21.7			
Normal (<15 ng/mL)	39	51.4	211	86.8	213	78.3			
(Ins) after 1 h OGTT	*n*	%	*n*	%	*n*	%	0.015	0.076	0.831
Elevated	6	7.9	3	1.2	14	5.1			
Normal	70	92.1	240	98.8	258	94.9			
(Ins) after 2 h OGTT	*n*	%	*n*	%	*n*	%			
Elevated	23	30.3	19	7.9	39	14.3			
Normal	53	69.7	224	92.1	233	85.7			

*p* < 0.05 Kruskal–Wallis test analysis.

**Table 5 ijerph-18-01060-t005:** Glucose concentration in study population during oral glucose test.

Glucose concentration	MetS+		MetS−		MetS+/−		MetS+ vs. MetS− vs. MetS+/−	MetS+ and MetS+/− vs. MetS−	MetS+ vs. MetS− and MetS+/−
**Fasting**	***n***	**%**	***n***	**%**	***n***	**%**	***p*-Value**		
<100 mg/dL	50	65.8	243	100	244	89.7	0.001	0.001	0.001
100–125 mg/dL	21	27.6	0	0	27	9.9	0.015	0.001	0.015
≥126 mg/dL	5	6.6	0	0	1	0.4	0.015	0.015	0.010
>200 mgL	0	0	0	0	0	0			
One hour after glucose intake									
Norm	75	98.7	243	100	272	100	0.985	0.985	0.961
>200 mg/dL	1	1.3	0	0	0	0			
Two hours after glucose intake									
Normal < 140 mg/dL	5	6.7	5	2.1	13	4.7	0.201	0.350	0.148
Elevated 140–199 mg/dL	71	93.3	238	97.9	259	95.3			
>200 mg/dL	0	0	0	0	0	0			

*p* < 0.05 Kruskal–Wallis test analysis.

**Table 6 ijerph-18-01060-t006:** HOMA-IR values (normal and elevated) in study subgroups.

HOMA-IR	MetS+		MetS−		MetS+/−		MetS+ vs. MetS− vs. MetS+/−	MetS+ and MetS+/− Vs. +MetS−	MetS+ Vs. MetS− and MetS+/−
	***n***	**%**	***n***	**%**	***n***	**%**	***p*-Value**		
Over ≥ 2.5	58	76.3	32	13.2	86	31.7	0.005	0.009	0.001
Normal i.e., <2.5	18	23.7	211	86.8	186	68.3			

*p* < 0.05 Kruskal-Wallis test analysis.

## Data Availability

Availability of data and materials: the data that support the findings of this study are available] but restrictions apply to the availability of these data, which were used under license for the current study, and so are not publicly available. Data are, however, available from the authors upon reasonable request and with permission of “6–10–14 for Health” integrated weight management program for children with overweight and obesity from Gdansk, Poland.

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
