# Peer review of "Metabolic Syndrome in Obese Children—Clinical Prevalence and Risk Factors"

_ijerph, 2021, doi:10.3390/ijerph18031060_

Round 1

Reviewer 1 Report

Thank you for the opportunity to review this paper.

It builds on previous work conducted by this group in the population of interest: “6–10-14 for Health”. Good article: Metabolic Syndrome in obese children – clinical prevalence and risk factors. The study aims to indicate predisposing factors for metabolic syndrome, especially those which can be modified. However, I have some suggestions for the improvement of the quality of the article to make it publishable. I have provided some general major comments below that should also be considered when reviewing.

-Abstract: In the section you should delete the sentence "…. Besides certain socioeconomic effects could be achieved too…." Because this study does not analyze the socioeconomic problems of metabolic syndrome” …. is inappropriate. Please revise the abstract as journal guidelines. It should be a single paragraph and should follow the style of structured abstracts.

-Materials and Methods: Please write the specific design. Please include the date and code register number of ethics committee.

- Results: Please, review the numbers in all tables to ensure they are correct, I am having trouble following. Additionally, remove the comma and replace with a period for the decimal place (as this is standard for this journal). Add 'by sex' to the foot notes for table 2 to clarify the statistical comparison.

-Discussion: Please revise the discussion and limitations. It's better to write the strength and limitation of the study as part of the discussion as per the journal guidelines.

Author Response

Please see the attachment."

Reviewer 2 Report

Review: “Metabolic Syndrome in obese children – clinical prevalence and risk factors.”

Overall, this is a good research project; highly practical and relevant; professional design, data collection, analyses and overall methodology; adequate sampling and so forth. The research and ‘cleaned up’ future version of the paper is certainly worthy of publication. More importantly, it is highly relevant for helping solve a growing ‘real world’ problem in many societies. I commend the research team for identifying this ‘problem’ (not necessarily obesity in general which has been known as an issue for some time, but the various sub-aspects involved which are often overlooked or missed); developing a professional and appropriate research approach to address it; and taking appropriate action.

However, the paper needs editing for publication in English (mainly word choice, grammar and some sentence structuring; the paragraphs, content and flow are adequate). Additionally, the tables can use some ‘cosmetic’ editing for better and consistent presentation and consistency (justification; cell/row/column consistency including the use of commas or periods for decimal points – table 1 and table 2 use commas and periods respectively… minor points but good for professional publication). Also, it seems (in my copy) that there are font and line spacing inconsistencies.

More importantly, some additional considerations and input concerning interpretations, speculations, impacts, recommendations, etc. may strengthen the paper as well. I am considerably interested in the cultural and behavioral aspects that may be at least partially “causal” and “can” be modified. Personally, I was rather impressed and certainly intrigued. I would definitely be interested in an expanded version with more author input, interpretation and speculations.

[note: I received only one supplementary file with tables – can they be inserted as an appendix to the article? Also, it would be useful to include the questionnaires/interview protocol (?) on diet, exercise, lifestyle, behavioral, etc. information that was collected.]

Many of the following statements are comments, suggestions, and questions. The authors are at liberty to use, modify or ignore these at their discretion. I may have misinterpreted some areas; erroneously assumed certain lines of logic in the methods, analysis and interpretations at some points; etc. Please ignore and forgive if this is the case.

Abstract:

Suggest removing the bullets. Separate paragraphs will suffice, or, can combine with introductory statements to each aspect. Also, it can be streamlined.

            The prevalence of childhood obesity

The prevalence of childhood obesity increases worldwide. The aim of this study was to indicate predisposing factors for metabolic syndrome (MetS), especially those factors that can be modified. The study comprised 591 obese children aged 10-12 years. Clinical examination, anthropometry, biometric impedance analysis, blood works (incl. OGGT and insulinemia) as well as dietary evaluation and physical activity were estimated (note: estimated or conducted ????). The risk factors of MetS or any of its component are: male; parents’ obesity (especially paternal as noted by authors in Results Section: “Further analyses revealed that 47.3% of fathers of children with MetS were obese and 14 5% fathers had normal body mass. Mother’s BMI did not differ between the groups.”); low body mass at birth; and omitting breakfast or dinner. There are key risk factors of metabolic syndrome in obese subjects. Some of these predictors can me modified, especially those regarding lifestyle. Identifying and then influencing these factors may help to reduce development of metabolic syndrome and consequently improve health and quality of life. Besides certain socioeconomic effects could be achieved too. (note to authors: this last statement is unclear).

Note: Need to clarify subject population’s ethnic, cultural, socio-economic, national, etc. backgrounds and other relevant demographic details. This is also an important factor in obesity issues. Poor suburban or rural southern US Americans, for example, may be quite different from affluent urban Central Asians for significantly different reasons. Only later in the text (Materials and Methods) can it be assumed the sample population is Polish.

Introduction:

Many statements are unclear or difficult to read without re-reading. They can be further clarified. Also, editing is needed throughout.

Example:

Prevalence of obesity is rapidly increasing. According to World Health Organization (WHO) in 2016 it affected as many as 650 million people and 2 billion adults were overweight.

The prevalence of obesity is rapidly increasing worldwide. According to the World Health Organization (WHO), it affected as many as 650 million people; two billion adults were also considered overweight (i.e., high risk for becoming obese); 41 million children younger than 5yrs and 340 million children or adolescents aged 5-19yrs were considered overweight or obese [1]. The number of children with obesity continues to increase and it is therefore expected that complications from obesity in these age groups will also increase.

Material and methods:

Not sure why there is a font change.

How thorough and accurate is the lifestyle and diet data? “Nutritional habits were evaluated by the dietitian on the basis of data given by children and parents. Special attention was paid on breakfast (first course within 2hr after wake up) and dinner (last meal eaten 2 hr before sleep).” – Is there an estimate on the accuracy? For example, direct observation is more accurate than 24 hour recall versus weekly recall estimates… The constraints for obtaining 100% accurate data are understood, but the authors may want to assure the reader that there is an estimate 80% or 90% accuracy (it more effectively legitimizes the data; but be honest to more effectively legitimize the researchers). “Special attention was paid on breakfast (first course within 2hr after wake up) and dinner (last meal eaten 2 hr before sleep).” – this is good, but the method need further clarification… maybe a few more sentences.

Statistical Analysis:

Line spacing (switch to double ?) is inconsistent.

References needed for Shapiro-Wilk, Mann-Whitney U, and ANOVA Kruskal-Wallis tests.

Stats analyses seem adequate for the study. Of course, I imagine there will be debate among other researchers concerning which statistical approaches are best (as is always the case), but having done many kinds of stats analyses and teaching stats for 30 years, I’m satisfied and confident the results are meaningful.

Results:

“Obesity was diagnosed in 85 (30,9%) girls and in 105 (33,2%) boys whereas, 190 (69,1%) girls and 211 (66,82%) boys were overweight. Girls had higher BMI centile compared to boys (p<0,0001).” – Do the 69.1% and 66.8% (omit the ‘2’ [100ths category] – be consistent) include or exclude the obese children? Need to clarify.

“Further analyses revealed that 47.3% of fathers of children with MetS were obese and 14.5% [note: include decimal – check for consistency and edit accordingly throughout document] fathers had normal body mass. Mother’s BMI did not differ between the groups.” – Thus, how much is possibly attributed to genetics and how much attributed to socialization and “family example” are influential of eating and exercise habits by parents, especially Fathers, and/or, even current cultural and behavioral habits including other children, family members, school, and so forth… although it is understood that this may be very difficult to test, determine, or even estimate. Nevertheless, speculations from authors would be interesting (just a few sentences). This is partially addressed by authors later on, “Both or either parent’s obesity - especially father’s - might depict genetic predisposition to excess body mass in a child, but, on the other hand, illustrates family’s lifestyle.” Nonetheless, a similar statement here might be appropriate as well.

“The correlation between metabolic syndrome and it’s compounds was significant for:

Father’s obesity (p=0.023)

Obesity in at least one parent (p=0.046)

Low body mass at birth in boys (p=0.046)”

Interesting. Good concise summary of main correlations. Is this consistent in other populations (e.g., as mentioned later by authors, “These results are similar to other publications [31].”? Any speculations on why? No need to answer if the data is unavailable, but a few ‘educated guesses’ would be interesting. I have my hypotheses, but would like to hear other hypotheses from other specialists more qualified than myself. Food for thought. Some of these issues and others are appropriately addressed in the following section: Nutritional analysis… especially eating habits of boys. How did they develop these habits?

Biochemical Assessment:

Good, good summary tables. Note: percentages are presented with periods in tables here, but commas in tables above – consistency.

Discussion:

“Apart from various definitions of MetS in children, apparently not all teens with obesity develop metabolic syndrome or not even one of its components. Which of the obese children (in fact whole obese population) are especially predisposed to MetS is still not fully understood. Yet, knowing the risk factors of metabolic syndrome would allow to prevent the latter or at least minimise its prevalence and consequences.” – Excellent point in final sentence; very relevant; very important! Something to include in abstract perhaps and in final statements.

“soon as possible to stop his process..” – I think one “.” Is sufficient (i.e., editing needed).

“…deep believe that…” deep belief – editing.

“…that obese children more often skip breakfast 34].” That obese children more often skip breakfast [34] – editing.

“…Obviously, there is not much we can do about genetics.” LOL – I do this a lot in my writings as well, but a more sterile ‘science – speak’ statement is likely in order. Suggested: “There is little that can be done about genetic predispositions and consequential effects.”

“…indicate modifiable risk factors…” Missing a period ‘.’ Here; maybe can borrow from above. Same here, “are required in order to lessen morbidity of obesity and metabolic syndrome” – missing period.

Conclusion:

I would recommend expanding the conclusion including further recommendations to address the problems; and future research recommendations.

Lastly, were there any significant constraints or shortcomings in the research? It is important to note, not to diminish the research, but to help this team and others for future research.

References:

Appears adequate.
